# The Evaluation of Geographical Health Facilities Structure in Makassar City, Indonesia

**DOI:** 10.3390/ijerph20065210

**Published:** 2023-03-16

**Authors:** Adipandang Yudono, Firman Afrianto, Annisa Dira Hariyanto

**Affiliations:** 1Department of Urban and Regional Planning, Universitas Brawijaya, Malang 65145, Indonesia; 2Department of Urban and Regional Planning, Universitas Gadjah Mada, Yogyakarta 55281, Indonesia; 3PT. Sagamartha Ultima, Malang 65142, Indonesia

**Keywords:** geographical structure, health facilities, big data, space syntax, spatial model, accessibility

## Abstract

Cities across the world, during the last period, have been shocked by the outbreak of the COVID-19 pandemic. The world of planning has since persevered in providing a response, in terms of how to anticipate this outbreak in the future. Various kinds of concepts have been issued, with various views and points of view. However, one of the needs for this planning is an appropriate evaluation of the geographic structure of existing health facilities, in order to properly provide consideration for future urban planning. This study attempts to provide an integrated model of how to evaluate the geographic structure of health facilities with a case study in Makassar City, Indonesia. By combining big data and spatial analysis, it is expected that it will find patterns and directions for acceptable health facilities planning.

## 1. Introduction

Cities across the world are currently experiencing a high level of urbanization and migration; this activity has the effect of producing an immense population densities as well. Several studies have stated that an increase in urban population has a positive influence on the urban economy, but it is also necessary to realize that an increase in population will directly put enormous pressure on urban space, one example of which is a decrease in environmental quality, which has an impact on public health [1,2,3,4,5,6]. The Indonesia National Commission on Human Rights has stated that health is fundamental for every individual, so it cannot be replaced for individual life; therefore health is recognized as a human right [7]. However, in reality there is still a question regarding how social protection is for every individual, to achieve access to feasible, optimal, and equal health facilities for various levels of society. In line with this crucial question, the researchers came to the same conclusion as Jausovec et al.’s statement, that the existence of health facilities that are optimally and evenly spatially distributed will allow each individual or layer of society to access health facilities [8]. The location of health facilities is a key factor that is closely related to the level of public accessibility to health services, and choosing the right location is also related to the success of the government’s health-related preventive programs. Based on these various considerations, the location of health facilities must be determined and evaluated appropriately.

COVID-19 has shaken society and disrupted all urban systems worldwide. This pandemic has also created significant global social changes, one of which is the restriction of public space utilization and social distance as the main policy made to reduce the spread of COVID-19 [8]. The COVID-19 pandemic in the context of regional and urban planning has presented seven main challenges in the fields of economy, transportation, urban density, public space, food security, commercial facilities, and social institutions (such as education and healthcare facilities) [9]. Unlike various other urban social facilities or institutions that are restricted during the COVID-19 pandemic, healthcare facilities are the only facilities that experience an increase in demand or utilization. COVID-19 has revealed significant deficiencies in healthcare facilities. However, of course, this demand trend has changed, where the need for healthcare facilities that are preferred is those closer to people’s residences. Based on the changes experienced after COVID-19, this requires us to redesign urban systems worldwide by drawing lessons from the history of COVID-19 and conducting significant reforms, especially regarding the planning of healthcare facility locations [10].

Many methods can be used and have been developed to solve the problem of choosing the location of health facilities; generally various studies use multi-criteria decision-making methods such as Spatial Multi-Criteria Evaluation [6], TOPSIS [11], or Fuzzy AHP/ANP [12]. As the selection of a health facility location is a problem with various criteria (environmental, social, economic, and political), it is considered a multi-criteria decision-making problem that can also be solved using a multi-criteria decision-making method. Besides this, significant technological developments have also allowed various studies to utilize the potential of using Geographical Information Systems (GIS) to improve the performance of the effective location selection of health facilities, providing benefits such as the ease of identifying underserved and fully-served areas [13].

Many complex factors determine the location of an effective health facility, including population, existing service levels, and current and future demand, and these must be carefully considered [14]. Of the various criteria for determining the location of health facilities, surveys have shown that the distance to health facilities is a key factor for the community in choosing health facilities [15]. Later, Gulliford et al., stated that access to health facilities is generally determined geographically, economically, or culturally [16]. Oppio et al. deepened the criteria for determining the location of facilities on a macro basis, in which the decision-making problem in selecting a location can be divided into four criteria [17]:Functional quality, considering the location as a city centre, its size and flexibility and building density;Site quality, considering access to the site, proximity, connection to green areas and sewerage network;Environmental quality, considering a quiet environment and avoiding air pollution;Economic aspects, considering land value, land ownership, and urban spatial planning.

From the various criteria mentioned, not many researchers consider spatial configuration in determining the location of health facilities. Haq & Luo found 24 quality articles that used space syntax (a method of assessing space configuration) in research related to health facilities, but were limited to assessing behaviour or movement (way finding) within the health facility area [18].

Space syntax is a new method for scientific modelling of cities, which has resulted in new scientific theories of cities [19]. The new scientific theory that Hillier meant is that cities must be planned and designed in a new way, that is, space is formed by networks. This indicates that we cannot only view cities as collections of buildings, but that they are even more complex, with networks, be it roads, activities, movements, and others. Based on these facts, the researchers hypothesize that in choosing the location of a health facility, it is also necessary to consider the spatial configuration, i.e., the interaction between the building and the surrounding network. The results of the spatial configuration with space syntax in this study will then be compared with the service radius (using the 5- and 10-min city concepts) and also public perceptions based on social media.

Along with the COVID-19 pandemic, the need for a safer and healthier environment has become increasingly crucial. The concept of the 5, 10, or 15-min city is considered an effective solution to meet this need. In this concept, all facilities and services needed by residents, such as workplaces, shopping centres, fitness centres, and healthcare facilities, are available within a 10 or 15-min distance from their residence [20]. Adequate and easily accessible healthcare facilities should be a priority in future settlement planning. Facilities such as health clinics and hospitals should be available in close proximity to every settlement. This way, residents do not need to travel far to meet their needs, thus reducing mobility and the risk of virus transmission. This concept has been adopted by several cities worldwide as part of their strategy to address the COVID-19 pandemic and create a healthier and more sustainable environment. The researchers believe that this study will be the first to combine the concept of healthcare service coverage equality with social world perceptions and service placement to evaluate the geographic structure of healthcare facilities, in which all data utilized Big Data.

Makassar City is one of the metropolitan cities (Mamminasata) in Indonesia and at the same time is the capital of the province of South Sulawesi. Makassar City is the fifth largest city in Indonesia and the largest in Eastern Indonesia [21]. The development and growth of Makassar City in its position as a core city in the Mamminasata Metropolitan area, is marked by the process of urbanization and rural-urban migration that is taking place very intensively [22]. As a centre for service in Eastern Indonesia, Makassar City plays a role as a centre for trade and services, a centre for industrial activities, a centre for government activities, a hub for goods and passenger transportation services both land, sea and air and a centre for education and health services. With the large role and function of the Makassar city, this study attempts to evaluate the geographic structure of health facilities in Makassar City so that an assessment of the effectiveness of the location of health facilities can be produced. In sustainable urban planning, finding the optimal and effective location for a facility has a crucial point for future land management [23].

## 2. Literature Reviews

### 2.1. Historical Pandemics and towards Healthy Cities Planning

Coronavirus disease 19 (COVID-19) is not only the first public health pandemic globally, though there are other pandemics that have hit the world [24]. This health pandemic has not only affected the health sector but has also caused urban problems and economic consequences. Throughout world history, cities have always been the origin of contagion. Several pandemic events originating in the city, namely the Black Plague in the 14th century, caused human deaths as far as Europe and the Middle East. Then, the Cholera outbreak in the 19th century period that caused the suffering of urban residents in London, Hamburg, Paris, New York, Chicago, and Moscow. Furthermore, the Great Flu epidemic in the era of 1918–1920, which caused the death of up to 50 million people worldwide. This major flu epidemic caused mass death of residents in the United States in cities such as Pittsburgh, Philadelphia, Louisville, and Nashville [25].

A health pandemic is the worst phenomenon when it spreads outside national borders. When an epidemic occurs involving respiratory elements, preventive actions are taken in the form of environmental isolation and closure of public spaces. Furthermore, this pandemic will change the image of cities and public spaces so that there are no community activities at all, but when the pandemic ends; will require changes in urban structure in integrating settlements with public health practices towards accessibility to health facilities in urban planning.

### 2.2. Healthy City

Healthy City have an old and a new concepts. The concept is “old” in the sense that humans have been trying to make cities healthier since the beginning of urban civilization. The “new” is related to in its manifestation as a main tool for health promotion of new public health in the search for Health for All [26]. In 1986, the Healthy Cities project was officially launched by the World Health Organization (WHO). The project is seen as a means to legitimize, maintain, and support the process of community empowerment [27,28]. Using community participation methods, this project seeks to reduce inequality, strengthen health gains, and reduce morbidity and mortality. The project’s goals are fairly conventional, but its methods and philosophy mark a defining shift in how to think about health in an urban environment. In 1991, the Healthy Cities Conference was held in Glasgow, England. The main point of the meeting was the recognition that many social scientists and epidemiologists are conducting research that addresses the various principles of Health for All, namely equality and inequality, participation, social change, public policy, healthy environment, individual skills development, and health service delivery issues.

The starting point for the Healthy Cities project was the recognition that cities have a meaningful role to play in promoting health and are in a unique position to implement public health initiatives. This is reflected in the latest thinking about ecology and the environment [28]. Furthermore, it is stated that the concept of Healthy Cities is a recipe for a quality life in a specific environment called the urban environment. According to the Healthy Cities philosophy, cities should provide a clean and safe physical environment based on a sustainable ecosystem [29]. Cities should offer citizens access to the prerequisites for health including food, income, shelter, and diverse experiences based on a diverse, important, and innovative economy. This has to happen against the background of various historical and cultural factors that are locally specific. Disseminating an understanding of the Healthy Cities philosophy is a foundational goal of the WHO projects.

Before discussing the Healthy Cities movement further, firstly, lets discuss some of the main elements that absolutely exist in the concept of Healthy Cities, e.g., how these various elements are applied, and the implications of these elements in research. Various conceptual elements are distinguished into two broad categories including the concept of health and the concept of how to obtain it. The three key elements related to health are a positive healthy model, an ecological healthy model, and attention to health inequalities. The main elements concern the strategy, which focuses on process; public policy; and community empowerment.

The Healthy Cities Project should be seen and understood within the health promotion context defined in the Ottawa Charter. Health promotion is the process of empowering people to increase control and improve health. According to the Ottawa Charter, health should be understood as a resource for everyday life that can help a person or group to identify and realize, to meet needs, and change with the environment. The Ottawa Charter emphasizes advocacy, enabling and mediation processes, as well as strategies for building health-informed public policies, creating an enabling environment, strengthening community action, developing individual skills, and reorienting health services. The Healthy Cities Project was envisioned as a tool to take these broad concepts and strategies and apply them at the local level [30,31].

There are three aspects of the healthy concept that are implicit and, to some extent explicit, in the health promotion and Healthy Cities models. First, being healthy as a positive concept does not only mean the absence of disease. Second, the holistic or ecological model of health takes into account all the different factors that affect health. Third, special attention to inequalities in health.

### 2.3. A Positive Health Model

The positive health model is rooted in the WHO definition that health is a state of complete physical, mental, and social health and not merely limited to the absence of disease or disability. This means that any measure of health, on an individual or a group basis, is not based solely on measures of mortality and morbidity [32]. This has long been recognized by public health circles, but in practice assessments based on measurements of life expectancy and infant mortality are still carried out in countries and cities.

Measuring morbidity is more difficult although there are few standardized measures of implementation such as health expectancy. Even less relevant measures such as the ratio of doctors or hospital beds to population are still frequently cited in city ranking publications.

One of the challenges is the need to educate the public and politicians that there is something more important to health than death and disease, that doctors and hospitals are not the main determinants of health status. The health of a city and its inhabitants needs to be measured in terms of physical, mental, and social well-being or fitness, with the same degree of importance as mortality and morbidity. However, there is no broad agreement on these measures.

### 2.4. An Ecological Health Model

An ecological healthy model or socio-ecological healthy model is also the basis of health promotion and the concept of Healthy Cities. This model recognizes that the determinants of health are multifactorial, incorporating various physical and social environmental determinants starting at the individual and global ecosystem levels. It is clear that as a model, health determinants go beyond the provision of hospitals and medical services. A wide range of public policies determine health, including various aspects of private company practice that impact the public at national and local levels. It is acknowledged that the central idea is that local government can and should play an important role in improving health and well-being.

The implications of the socio-ecological model of health for community research include the development of salutogenic epidemiology, which is a good cause of health epidemiology that can also measure the relative strength of different determinants of health, and the projected or actual impact of salutogenic interventions on health. The term salutogenic is taken from Antonovsky [33], a health sociologist who seeks to answer why some individuals can live relatively well in situations where others do not. He was then interested in salutogenesis, which is the opposite of pathogenesis, which was already known in the medical field.

### 2.5. Health Inequalities

As a strategy used to achieve Health for All, health promotion pays attention to inequalities in health and various efforts to improve health for the unhealthiest groups in society. This is because inequality in health is rooted in injustice in accessing basic health conditions; health promotion pays attention to social injustice. These characteristics are the same as those of the Healthy Cities project. Dealing with inequities in health, a number of urban problems emerge. First, is there sufficient data at the city level to document inequalities in health within the city? Second, what is more important is whether cities have the power and jurisdiction to implement various measures that address inequity in accessing basic determinants of health.

Based on this understanding of healthy cities, this study will explore a model that integrates geographical structure with basic access to health facilities. In advance, the research introduces the use of Big Data and spatial analysis in finding patterns and directions for planning health facilities.

### 2.6. The Walkability Performance to Health Facilities

The 15-min city concept is not the newest study in the world of planning [34]. Many planning scholars who have conducted research on inclusive cities have adopted the 15-min city, which, in general can be divided into two study groups, namely neigbourhood units and accessibility.

The concept of a neighbourhood unit as a structural unit of an organized city was introduced in the 1920s by Clarence Perry. Perry conducted neighbourhood unit studies with exploration as part of a city and a community entity in urban settlements [35]. Furthermore, the neighbourhood unit proposed by Perry sets design standards in the form of dimensional factors (e.g., providing 10% creative space and a garden in a residential community entity dwelling) equipped with functional facilities, such as public services, shops, schools, offices).

Furthermore, in the 1950s, Vincenzo Columbo, a planning scholar from Italy created a vision of urban structure known as organic urban planning. The concept of organic urban planning divides three types of urban structure hierarchies, namely neighbourhoods, districts, and communities [36]. The division of the type of city structure is based on the daily movement of the population that is oriented towards walking and then innovating to the concept of the 15-min city.

In the subsequent development of the urban concept, the traditional structure of neighbourhood units was adopted by Calthorpe, which is called new urbanism [37]. The new urbanism proposed by Calthorpe connects residential units with settlement- supporting facilities and infrastructure (markets, schools, offices, public services, etc) by minimizing the mobility distance that can be reached on foot. Furthermore, the new concept of urbanism introduced by Calthorpe is known as the Transit-Oriented Development (TOD) model, which is characterized by mixed land uses and varying settlement densities so that all of these land-use functions can be reached on foot within an ideal travel time, i.e., 15 min.

The TOD model, which introduces connectivity between land uses within 15 min, was also adopted by Mezzoued et al. (2021) [38] with the concept of a slow city, namely creating connectedness between land uses through slow mobility via walking, where the ideal concept of walking time to move from one functional to another functional environment is 15 min.

The real concept of the 15-min city was actually expressed by Moreno et al., which combines the concepts of environment and accessibility units [20]. This city concept introduces chrono-urbanism, as a city that is resilient, sustainable, and inclusive. Furthermore, this 15-min city concept translates into the Urban Agenda 2030, namely: decentralizing facility and infrastructure services at the core of the environment with non-rigid usage, involving the community in urban planning and increasing cycling and walking infrastructure development.

The 15-min city study introduces land use connectivity with each other, which can be reached on foot in an ideal travel time of 15 min. Previous studies examined general concepts in urban planning to create an ideal urban concept, however, related to the COVID-19 pandemic that has hit the world, the problem of healthy urban planning, in order to realize resilient, resilient and sustainable cities, has still not been answered. For this reason, to fill the gaps in the lack of research in the healthy city concept, this study explores specific studies regarding the attachment of public health facilities to residential environments under 15-min travel time.

## 3. Methods

### 3.1. Data Sources

Several digital data sets containing spatial data were collected through various online sources or Big Data (Table 1): for regional boundaries, data from GADM.org levels 2 and 3 were used for the Indonesian Region and were deducted for Makassar City and its sub-districts; road network data were obtained from the Open Street Map; population data were obtained from the population raster data from the WorldPop site with a resolution of 100 m; and facility data were obtained from Google POI data, simplified into Hospital, Medical Clinic, Dental Clinic, Doctor, and Pharmacy facility classes.

### 3.2. Service Area Analysis

Service area analysis shows the coverage area of the facility based on the distance to a location. The results of this analysis are health service area polygons with the main principle of access varying with impedance [23]. One effective model that can describe service areas, which was created to help cities adapt to post-COVID realities is the “15-min city”. This concept was first initiated by Carlos Moreno in 2016 [20]. The core aim of this concept is that cities should be designed in such a way that, within walking or cycling distance (e.g., 15 min) from their homes, people can fulfil all their daily needs such as jobs, housing, food, health, education, culture, sports, and recreation. Referring to this concept, this study will use service area analysis using time contours of 5 and 10 min (5–10 min city concept) by driving a motorized vehicle with an average speed of 30 km/hour, supported by the Valhalla Plugin in QGIS. 3.22.

### 3.3. Density Analysis

Conventionally, there are three methods of density analysis that have been widely used in geographic related studies, which are quadratic analysis, Voronoi-based analysis, and kernel density method [39]. Density analysis in this study uses kernel density, which is a non-parametric estimation method used to estimate the density of the overall probability distribution without assuming the overall distribution function [40]. The following is the kernel density formula:$$f_{h}\left(x\right)=\frac{1}{nh}\overset{n}{\underset{i=1}{\sum }}K\left(\frac{x-x_{i}}{h}\right)$$
where *f_h_*(*x*) is the density of kernels, and the greater the value of *f_h_*(*x*), the higher the density of facilities at that location. The symbols *n* represents the total number of grids in the study area, *K*(*ÿ*) is a kernel function, *x_i_* is the observations that are distributed independently, and *x* is the average value of the observations [40].

### 3.4. Space Syntax

Space syntax is a theory and method that has developed over the last 40 years. It was initially conceived as a theory of society and space, but this method has since expanded to other fields [18]. The space syntax theory is a method for measuring the environment as a set of predictive variables for certain behaviours, which then predict the behaviour of walking in a public area or a facility by targeting the tendency of people to move to spaces with a higher level of integration [41].

Spatial syntax in this study will be assessed using four metrics from Spatial Design Network Analysis (SDNA), including: (1) Closeness Centrality; (2) Connectivity; (3) Network Gravity; and (4) Betweenness Centrality. SDNA is included in the ArcGIS/QGIS/AutoCAD and Python plugin sections for 2D and 3D network analysis.

Closeness centrality: It is defined as a way to measure centrality in a network that focuses on how much a node has a minimum distance from other nodes. The numbers displayed in the closeness algorithm are different from other syntax algorithms, which the smallest number indicates the highest concetration;Connectivity: This is usually calculated using the degree of centrality, where the greater the value, the more the edges connected by nodes [42]. Connectivity is also interpreted as a way to measure centrality in a network that focuses on how much a node is connected with other nodes;Network gravity or gravity index: It describes the number of other nodes that can be reached from the shortest path distance at most, also considering the two spatial directions needed to reach each destination [43];Betweenness centrality: The greater the value, the more the shortest path between pairs of different nodes must pass through these nodes [30,31,32,33,34,35,36,37,38,39,40,41,42]

### 3.5. Correlation

Correlation analysis is a statistical method used to determine the relationship between two variables. The goal is to determine whether there is a linear relationship between the two variables and, if so, how strong the relationship is. In this study, we wanted to examine the relationship between social world perception and space configuration for health facilities in Makassar City using Pearson’s correlation.

The research flowchart in this study is described in a structure in Figure 1:

## 4. Results and Discussion

Data preparation was done first, before proceeding to the next process. As discussed in the previous chapter, this study focuses on the geographic evaluation of a healthcare facility, where we will look at the Equalization of Service Coverage, Service Centralization, and Service Placement of healthcare facilities in a region. The administrative district area of *kecamatan* in Makassar City was obtained from GADM.org data. This administrative area was used to assess the level of equity. To obtain healthcare facilities according to the *kecamatan* administrative boundary, a zonal statistic process was conducted in QGIS. Healthcare facilities in Makassar city were obtained from Google POI using the keyword “health facility”. The results of this collection were then sorted by facility type and grouped into five major categories: Dental Clinic, Doctor, Hospital, Medical Clinic, and Pharmacy. User-generated review data were not removed and used as the basis for digital world perception of the facilities in question. Road network data, used as the basis for network analysis and spatial configuration analysis, were obtained from OSM. After all the data were obtained and processed, an analysis process was carried out, which will be explained in the following sub-chapter.

### 4.1. Health Facilities in Makassar City

Makassar City consists of 14 *kecamatan* (districts) with a total of 403 health facilities and a population (as of 2021) of 1,428,096 people (Table 2). The districts that have the highest facilities are Rappocini District, Panakkukang District, and Makassar District. Meanwhile, the districts with the highest population are Tamalanrea and Tamalate (Figure 2).

The number of health facilities by type in Makassar City consists of 177 doctors (44%), 120 hospitals (30%), 60 medical clinics (15%), 25 pharmacy units (6%), and 21 dental clinics units (5%) (Table 3, Figure 3).

### 4.2. Health Facility Service Range

The first evaluation regarding the equalization of healthcare facilities is using service area analysis. Service area analysis uses the assumption of driving range from the healthcare facility. The assumptions used include: (1) people use motorized vehicles to reach healthcare facilities, (2) motorized vehicles move at an average speed of 30 km/h, and (3) the route taken uses the road network available in OSM. In GIS calculation, the service area can be obtained in the form of a polygon, so it can be visualized in terms of how it compares to the total area of the city.

The range of health facility services in Makassar City, based on the latest post-pandemic city concept using 5- and 10-min time units, has been able to serve more than 80% of the population (Table 4, Figure 4).

In terms of the equalization of healthcare facilities according to the total number, it may already cover almost all areas of the city, but it is necessary to investigate the equalization according to the type of healthcare facilities. The discussion below will cover it according to time contour and facility type.

#### 4.2.1. Service Range of 5 min

A span of 5 min is able to serve 66.69% of the total area and 89.84% of the total population. The highest type of health facility capable of serving the total area is a hospitals (63.60%) and the lowest is a pharmacies (29.79%). The highest type of health facility capable of serving the total population is hospitals (87.90%) and the lowest is pharmacies (49.43%) (Table 5, Figure 5).

#### 4.2.2. Service Range of 10 min

A span of 10 min is able to serve 90.54% of the total area and 99.14% of the total population. The highest type of health facility capable of serving the total area is a hospitals (89.95%) and the lowest is pharmacies (51.57%). The highest type of health facility capable of serving the total population was hospitals (99.09%) and the lowest was pharmacies (72.53%) (Table 6, Figure 6).

### 4.3. Centralized Facility Services

To obtain an evaluation of Service Centralization, the analysis used is the point density analysis with the weight of “reviews” from Google Maps POI. Unlike the geographic analysis commonly used by researchers, which uses point density without any weight of whether a facility is attractive and visited frequently, the method of using “reviews” weight from Google Maps POI can provide a more realistic picture and can solve the difficulty of obtaining service data in a region.

The first step was to find where the centralization is according to the *kecamatan* in Makassar City. This method was done by conducting zonal statistics of the number of healthcare facilities into *kecamatan*/districts. The division according to natural breaks (jenk) is carried out to obtain a visualization of high to low.

Centralization of the highest health facilities is in *kecamatan* Rappocini. Google’s highest POI review in aggregate is in *kecamatan* Panakkukang, *kecamatan* Rappocini, and *kecamatan* Mariso. Meanwhile the average Google POI rating is the highest in aggregate in *kecamatan* Manggala, *kecamatan* Makassar, and *kecamatan* Ujung Tanah (Figure 7).

The second step was to calculate how centralization is according to its “reviews” weight through kernel density analysis. The result of this analysis is the area (raster) where the centralization of healthcare facilities is located. This visualization can be used as a basis for future planning to determine whether a new healthcare facility needs to be built on the suburb of the city or in other settlement centres. Another advantage that can be gained is that we can evaluate how people rate each facility, where it is conveyed in the range of 0–5 on Google Maps POI.

Social media perception of the highest number of Google POI reviews is hospital by 60% and the lowest is pharmacy by 2%. Meanwhile, from the average Google POI rating, the highest is a dental clinic at 23% and the lowest is a hospital at 17% (Table 7, Figure 8 and Figure 9).

### 4.4. Facility Placement Evaluation

Space configuration is the arrangement of space and its components, including buildings, roads, and public spaces. It is an important aspect of urban planning and design that can impact how people interact with their environment.

One method for analyzing space configuration is space syntax. Space syntax is a technique used to analyze the spatial properties of urban environments by measuring the accessibility, connectivity, and visibility of different spaces. It uses graph theory to model the spatial relationships between different spaces and identify the underlying patterns of movement and interaction. The space syntax approach involves mapping the urban environment into a graph-based representation of the street network. This representation is then analyzed using a set of metrics that capture different aspects of spatial configuration, such as the degree of connectivity, the level of integration, and the centrality of different spaces. These metrics can be used to identify areas of the urban environment that are more or less accessible, connected, or visible, and to determine how these properties affect pedestrian movement, social interaction, and other urban phenomena. By using space syntax, planners and designers can gain insights into the spatial properties of urban environments and develop strategies for improving accessibility, connectivity, and visibility. This can include interventions such as reconfiguring street networks, enhancing pedestrian routes, and creating more open and visible public spaces. Ultimately, the goal of space syntax is to create more livable, sustainable, and equitable urban environments that support the needs and aspirations of all residents.

Space syntax analysis can be used to identify the most central and accessible locations for healthcare facilities, based on factors such as the degree of connectivity to surrounding streets and the level of integration with other urban spaces. This analysis can also help identify areas of the city that are underserved by healthcare facilities and need new facilities to be established. In addition, space syntax can be used to evaluate the impact of health facility placement on urban mobility and accessibility. For instance, it can help identify pedestrian routes that are obstructed or poorly connected to healthcare facilities, and suggest interventions such as creating new pedestrian paths or improving existing ones to enhance mobility and accessibility. Overall, space syntax analysis is a useful tool for evaluating health facility placement and improving access to healthcare services in urban areas.

The algorithm produced by space syntax is a mathematical model that uses graph theory to analyze spatial configurations and their impact on urban mobility and accessibility. The algorithm calculates a set of metrics that capture different aspects of spatial configuration, such as the degree of connectivity, the level of integration, and the centrality of different spaces. The main concept behind the space syntax algorithm is that spatial configurations influence the way people move and interact in urban environments. By analyzing the spatial relationships between different spaces, the algorithm can identify patterns of movement and interaction and evaluate how these patterns are affected by changes in the spatial configuration. The algorithm produces a range of metrics, each of which provides insights into different aspects of spatial configuration. These metrics include:Closeness: This measures the extent to which a space is connected to other spaces in the urban network.Betweenness: This measures the number of alternative paths that can be taken to reach a particular space.Gravity: This measures the degree to which a space is strategically located within the urban network.Connectivity: This measures the importance of a space in terms of its position within the network.

By analyzing these metrics, the space syntax algorithm can identify the most important spaces in the urban network, such as central locations that are well-connected and highly visible. It can also identify areas that are underserved by urban amenities, such as health facilities, and suggest interventions to improve access and mobility.

In space syntax analysis, global configuration refers to the overall spatial structure and organization of an urban environment, while local configuration refers to the immediate surroundings of a specific location or space. The global configuration of an urban environment is analyzed using metrics such as the degree of connectivity, the level of integration, and the centrality of different spaces. These metrics capture the overall structure and organization of the urban network and can help identify important spatial features such as the most central locations or the most well-connected areas of the city.

On the other hand, local configuration refers to the immediate surroundings of a particular location or space, typically within a radius of 400 m. The analysis of local configuration can help identify how a specific location is connected to its immediate surroundings and how this affects movement and accessibility. For example, local configuration analysis can help identify pedestrian paths that are obstructed or poorly connected to surrounding areas, as well as areas that are underserved by urban amenities such as health facilities. This information can be used to suggest interventions such as creating new pedestrian paths, improving existing ones, or establishing new health facilities in underserved areas.

Overall, the analysis of both global and local configuration is important for understanding the spatial properties of urban environments and their impact on mobility and accessibility. by using metrics such as connectivity, integration, and centrality, space syntax analysis can help identify areas of the city that need improvement and suggest interventions to enhance mobility and accessibility for all residents. Global Spatial Configuration

#### 4.4.1. Global Spatial Configuration

In a global spatial review, the highest connectivity value was identified in medical clinics and the lowest in hospitals. The highest closeness value was identified in dental clinics and the lowest at hospitals. The highest gravity value was identified in pharmacies and the lowest in hospitals. The highest value of betweenness was identified at dental clinics and the lowest at the hospitals (Table 8, Figure 10).

The highest correlation between the number of reviews and global spatial configuration is in the gravity algorithm, which shows that the closer the facility to the centre of the city road network, the higher the number of reviews (public perception).

The highest correlation between the average rating and global spatial configuration is in the closeness algorithm, which shows that the closer the facility to the centre of the city road network, the higher the rating given (Table 9).

#### 4.4.2. 400-m Radius Area Configuration

In a review of the 400-m radius, the highest connectivity value was identified in medical clinics and the lowest in hospitals. The highest closeness value was identified in hospitals and the lowest in doctors. The highest gravity value was identified in pharmacies and the lowest in dental clinics. The highest value of betweenness was identified in medical clinics and the lowest in hospitals (Table 10, Figure 11).

The highest correlation between the number of reviews and the configuration of the 400-m radius area is in the connectivity and betweenness algorithm, which shows that the closer the facility to the centre of the local network and those on the local main network, the higher the number of reviews (public perception).

The highest correlation between the average rating and the configuration of the 400-m radius area is in the betweenness algorithm, which shows that if the facility is on the local main network, the higher the rating is given (Table 11).

## 5. Conclusions

The evaluation of geographic structure on health facility placement is crucial for ensuring that healthcare services are accessible and available to all members of a community. The geographic structure of an area can significantly impact the distribution and availability of healthcare facilities, as well as the accessibility of these facilities to different segments of the population. By conducting an evaluation of geographic structure on health facility placement, planners and policymakers can identify areas that are underserved by healthcare facilities and develop strategies for improving access and availability. This can include establishing new facilities in underserved areas, improving transportation infrastructure to connect patients to existing facilities, or developing mobile healthcare services to reach remote or rural areas.

Moreover, the evaluation of geographic structure can also help improve the efficiency and effectiveness of healthcare services. By analyzing the spatial distribution of healthcare facilities, planners and policymakers can identify areas of duplication or overlap and suggest consolidation or integration of facilities to improve service delivery and reduce costs. Overall, the evaluation of geographic structure on health facility placement is essential for improving the accessibility and availability of healthcare services and ensuring equitable access for all members of a community. By using tools such as GIS and space syntax analysis, planners and policymakers can make informed decisions that lead to more efficient and effective healthcare services and a healthier and more vibrant community.

In addition to the evaluation of geographic structure on health facility placement, the integration and use of Big Data can also be crucial for improving healthcare services and outcomes. Big Data refers to the large amounts of structured and unstructured data that are generated in various fields, including healthcare. This information can be used to develop targeted interventions to improve healthcare access and outcomes, as well as to reduce costs and improve the efficiency of healthcare service delivery.

Regarding the case discussed in this research, which is the health facilities in the city of Makassar, the integration and use of Big Data can provide valuable insights into the distribution and accessibility of healthcare services in the city. The evaluation of geographic structure on health facility placement can also provide important insights into the distribution and accessibility of healthcare services in the city. By using tools such as GIS and space syntax analysis, planners and policymakers can analyze the spatial relationships between health facilities and other urban spaces to determine how they are connected and how they affect people movement and access to healthcare. The distribution of health facilities in Makassar City is acceptable in terms of the service coverage radius. However, in the future planning, the city needs to focus on adding types of pharmaceutical health facilities. In terms of service concentration, health facilities in Makassar City rely on the city centre, so it is necessary to build health facilities in suburban areas, especially in the northern part of the city (Tamalanrea and Biring Kanaya) and the southern part of the city (Mariso and Tamalate). In terms of evaluating the placement, the future planning of health facilities needs to pay attention to spatial configuration, especially the placement of health facilities according to the connectivity and betweenness algorithms at a radius of 400 m to properly achieve the concept of the 10 min city and the neighborhood unit.

One potential weakness of this study’s reliance on Big Data is that the analysis is dependent on the availability and accuracy of the data. In some cases, the data may be incomplete, inaccurate, or outdated, which could lead to incorrect conclusions or recommendations. Additionally, the analysis may be biased towards certain demographic groups or geographic areas, depending on the data sources and the availability of data. Furthermore, the use of Big Data may also raise concerns about privacy and data security. The collection and analysis of large amounts of personal data may raise ethical concerns and may require safeguards to protect the privacy of individuals.

Another potential weakness of the study is that the use of space syntax analysis and other GIS-based methods can be complex and technical, requiring specialized knowledge and expertise to interpret and analyze the data. This may limit the ability of non-experts to fully understand and utilize the results of the study. Overall, while the use of Big Data and GIS-based analysis can provide valuable insights into the distribution and accessibility of healthcare services, it is important to be aware of the limitations and potential biases of these methods. To address these limitations, it is important to use multiple sources of data and to validate the results of the analysis through on-the-ground observations and community engagement.

## Figures and Tables

**Figure 1 ijerph-20-05210-f001:**
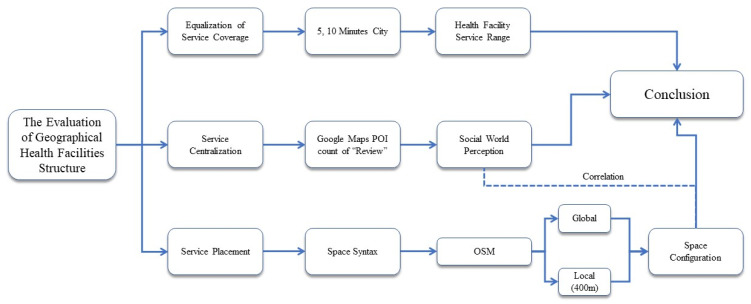
Research Flowchart.

**Figure 2 ijerph-20-05210-f002:**
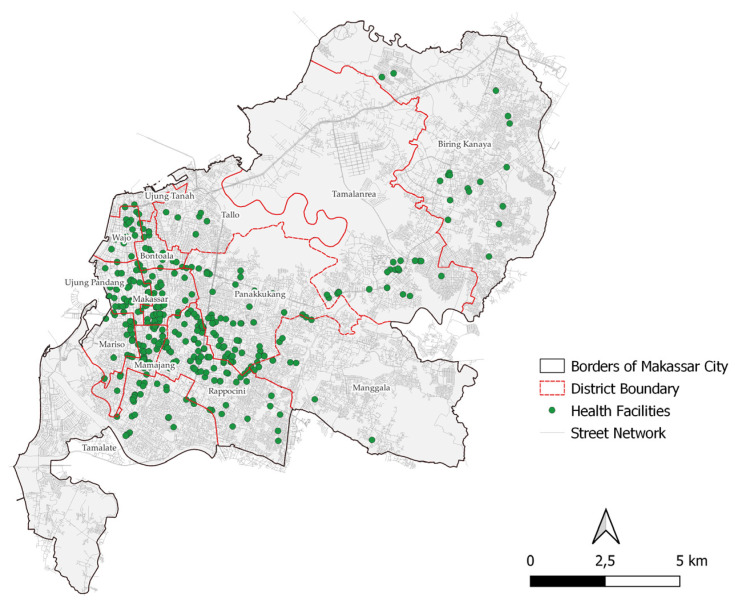
Distribution of Health Facilities in Makassar City.

**Figure 3 ijerph-20-05210-f003:**
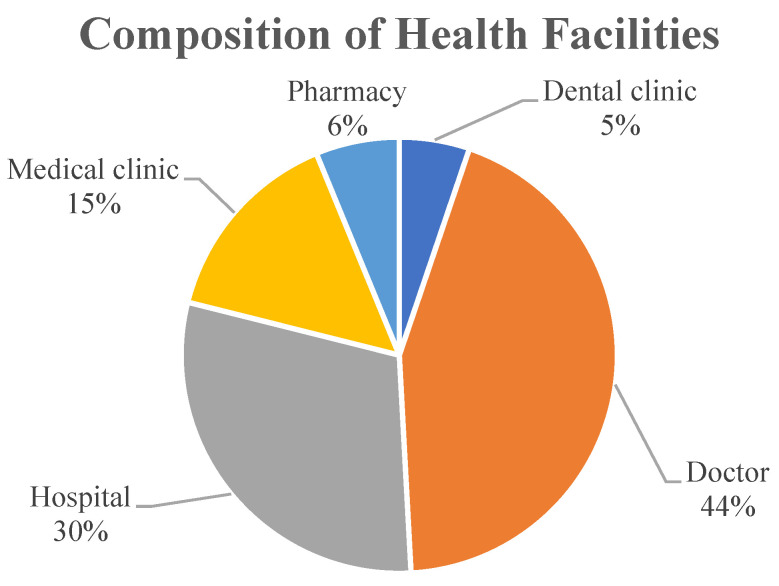
Composition of Health Facilities in Makassar City.

**Figure 4 ijerph-20-05210-f004:**
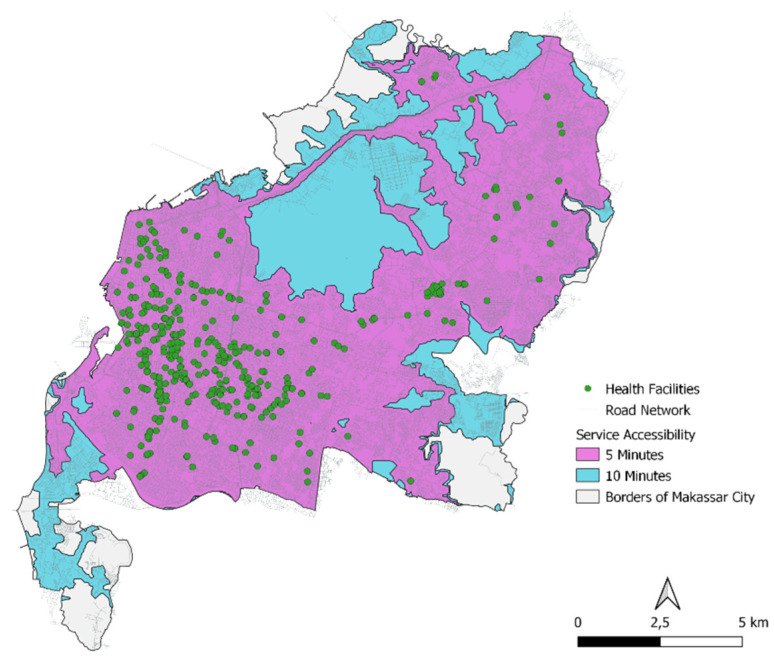
Outreach Map of Health Facilities according to Time Contour.

**Figure 5 ijerph-20-05210-f005:**
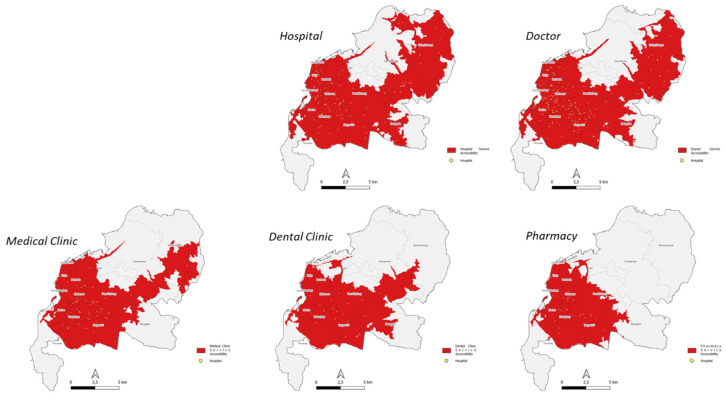
Health Facility Outreach Map According to a Time Contour of 5 min.

**Figure 6 ijerph-20-05210-f006:**
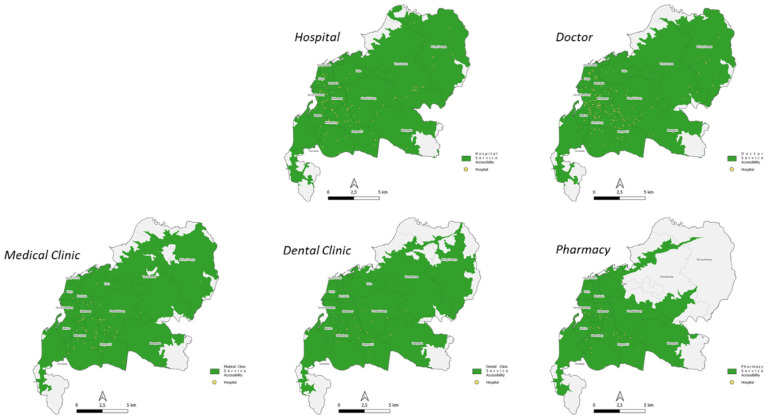
Health Facility Outreach Map According to Time Contour 10 min.

**Figure 7 ijerph-20-05210-f007:**
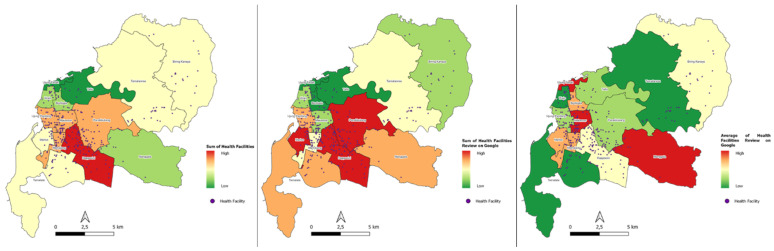
Map of Health Facility Service Concentration by Kecamatan administrative boundary.

**Figure 8 ijerph-20-05210-f008:**
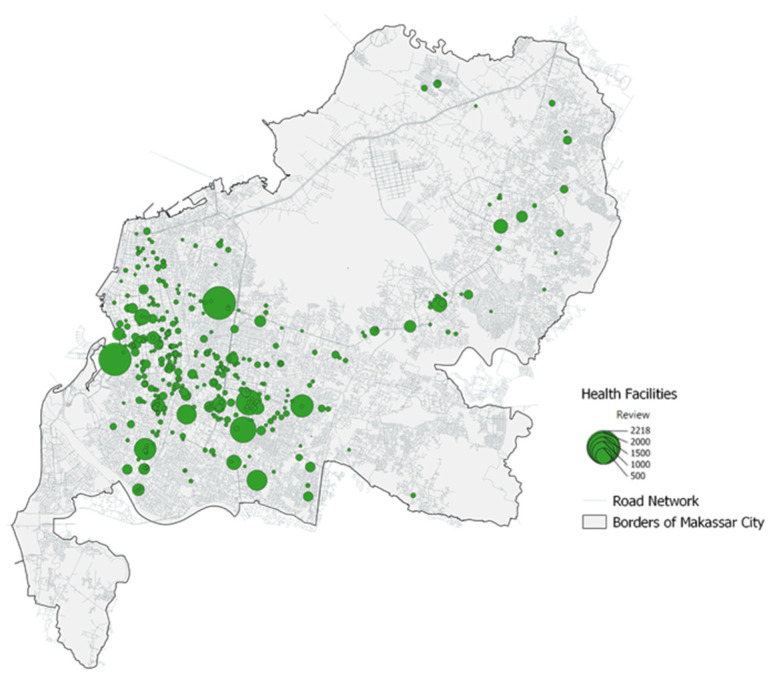
Map of Health Facility Service Perceptions according to Google POI.

**Figure 9 ijerph-20-05210-f009:**
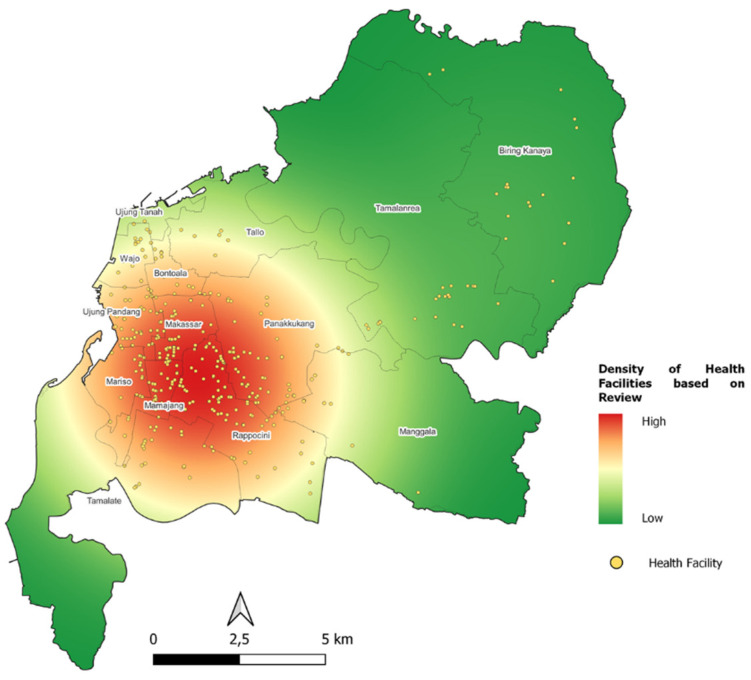
Health Facilities Kernel Density Map by Number of Google Reviews.

**Figure 10 ijerph-20-05210-f010:**
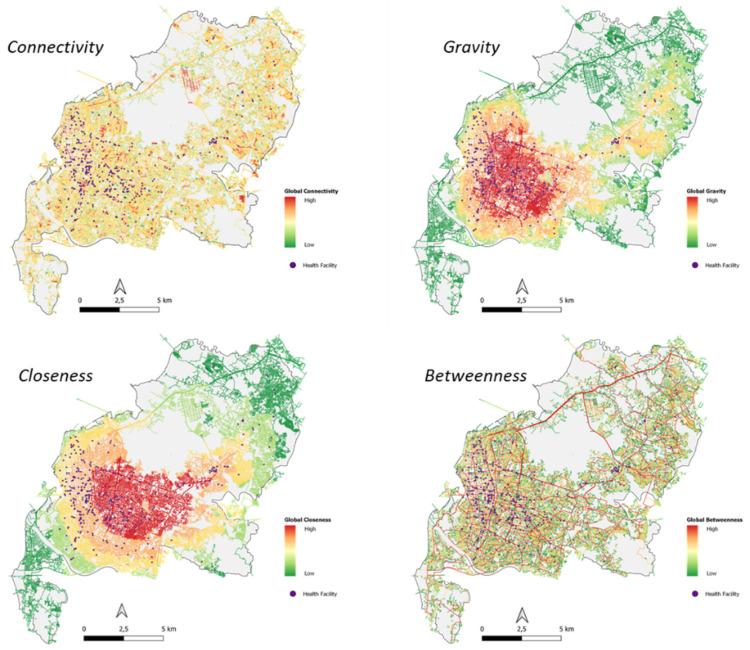
Global Spatial Review Configuration Map and Health Facility.

**Figure 11 ijerph-20-05210-f011:**
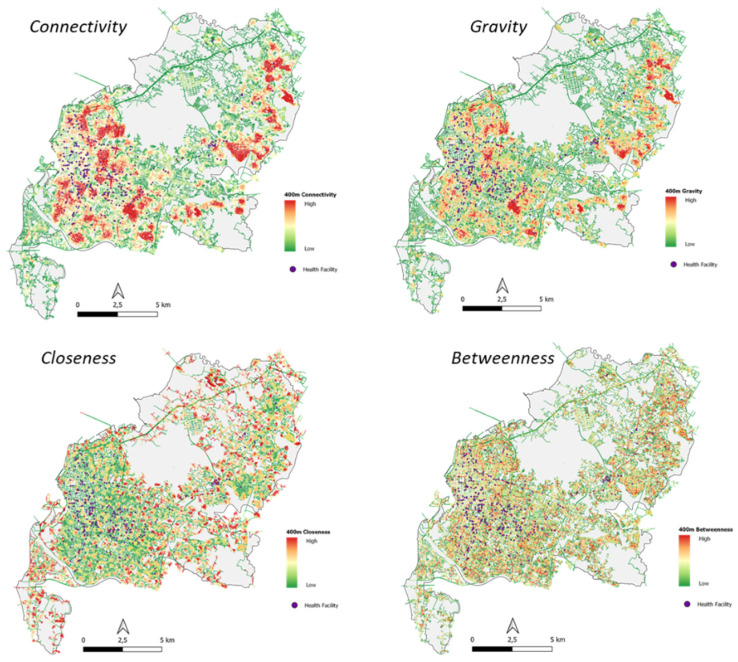
Configuration Map of 400-m Radius Area and Health Facility.

**Table 1 ijerph-20-05210-t001:** Research Dataset.

Data Type	Data Source
Administrative boundary	GADM.Org (Data Collection: December 2022)
Road network	OpenStreetMap (Data Collection: December 2022)
Population	WorldPop (Data Collection: December 2022)
Health Facilities	POI Google (Data Collection: December 2022)

**Table 2 ijerph-20-05210-t002:** Distribution of Health Facilities by Kecamatan.

District	Total Health Facilities	Total Population
Biring Kanaya	18	258,889
Bontoala	16	27,430
Makassar	44	42,227
Mamajang	41	38,321
Manggala	12	154,327
Mariso	20	33,374
Panakkukang	57	155,324
Rappocini	84	144,068
Tallo	6	78,090
Tamalanrea	22	211,474
Tamalate	24	195,388
Ujung Pandang	40	38,353
Ujung Tanah	2	19,091
Wajo	17	31,740
Grand Total	403	1,428,096

**Table 3 ijerph-20-05210-t003:** Distribution of Health Facilities in Makassar City.

Health Facility	Number by Type
Dental clinic	21
Doctor	177
Hospital	120
Medical clinic	60
Pharmacy	25
Grand Total	403

**Table 4 ijerph-20-05210-t004:** Health facilities Service Coverage according to Time Contour.

Time Contour (min)	Area (ha)	Population	% to Total Area	% to Total Population
10	15,783.89	1,416,300	90.4%	99.14%
5	11,625.83	1,283,357	66.69%	89.84%

**Table 5 ijerph-20-05210-t005:** Service Range of Health Facilities in 5-min Drive.

Time Contour	Type of Health Facility	Area (ha)	Population	% to Total Area	% to Total Population
5	Dental clinic	6868.94	885,084	39.40%	61.96%
5	Doctor	10,606.34	1,216,620	60.84%	85.16%
5	Medical clinic	7950.65	983,354	45.61%	68.84%
5	Hospital	11,087.84	1,255,649	63.60%	87.90%
5	Pharmacy	5194.18	706,165	29.79%	49.43%

**Table 6 ijerph-20-05210-t006:** Service Range of Health Facilities in 10-Minute Drive.

Time Contour	Type of Health Facility	Area (ha)	Population	% to Total Area	% to Total Population
10	Dental clinic	13,709.32	1,328,066	78.64%	92.96%
10	Doctor	15,340.29	1,407,338	87.99%	98.51%
10	Medical clinic	14,559.89	1,375,882	83.52%	96.31%
10	Hospital	15,682.14	1,415,599	89.95%	99.09%
10	Pharmacy	8990.98	1,036,095	51.57%	72.53%

**Table 7 ijerph-20-05210-t007:** Perceptions of Health Facility Services Based on POI Google.

Health Facility	Count of Type	Sum of Review	% Sum of Review	Average of Rating
Dental clinic	21	2805	11.89~12	4.19
Doctor	177	2438	10.34~10	3.22
Hospital	120	14,114	59.83~60	3.14
Medical clinic	60	3714	15.74~16	3.88
Pharmacy	25	518	2.20~2	3.66
Grand Total	403	23,589	100	3.37

**Table 8 ijerph-20-05210-t008:** Perception of Healthcare Facility Services and Global Spatial Review Configuration by Google POI.

Row Labels	Sum of Review	Average of Rating	Average of Connectivity	Average of Closeness	Average of Gravity	Average of Betweenness
Dental clinic	2805	4.19	4.10	7780.98	11.10	313.24
Doctor	2438	3.22	4.23	8248.65	10.68	215.19
Hospital	14,114	3.14	3.91	8490.87	10.26	125.74
Medical clinic	3714	3.88	4.35	8043.44	11.05	165.34
Pharmacy	518	3.66	4.24	8142.16	11.13	288.86
Grand Total	23,589	3.37	4.14	8259.25	10.66	190.81

**Table 9 ijerph-20-05210-t009:** Correlation of Digital Perception with Global Spatial Configuration.

	Sum of Review	Average of Rating	Average of Connectivity	Average of Closeness	Average of Gravity	Average of Betweenness
Sum of Review	1					
Average of Rating	−0.54	1				
Average of Connectivity	−0.80	0.37	1			
Average of Closeness	0.68 **	−0.95 *	−0.43	1		
Average of Gravity	−0.87 *	0.84 **	0.72	−0.86	1	
Average of Betweenness	−0.75	0.63	0.23	−0.74	0.74	1

* Negative Strong Correlation and significant. ** Positive Strong Correlation and significant.

**Table 10 ijerph-20-05210-t010:** Health Facility Service Perception and 400 m Radius Area Configuration according to Google POI.

Row Labels	Sum of Review	Average of Rating	Average of Connectivity	Average of Closeness	Average of Gravity	Average of Betweenness
Dental clinic	4.19	2805	185.05	263.80	0.49	2.31
Doctor	3.22	2438	200.38	269.75	0.50	2.44
Hospital	3.14	14,114	176.83	262.95	0.50	1.94
Medical clinic	3.88	3714	204.87	267.61	0.52	2.98
Pharmacy	3.66	518	199.00	266.18	0.55	2.93
Grand Total	3.37	23,589	193.15	266.87	0.51	2.40

**Table 11 ijerph-20-05210-t011:** Correlation of Digital Perception with 400-m Radius Area Configuration.

	Sum of Review	Average of Rating	Average of Connectivity	Average of Closeness	Average of Gravity	Average of Betweenness
Sum of Review	1					
Average of Rating	−0.54	1				
Average of Connectivity	0.17	−0.75	1			
Average of Closeness	−0.18	−0.59	0.88	1		
Average of Gravity	0.08	−0.48	0.58	0.28	1	
Average of Betweenness	0.42 **	−0.76 *	0.89	0.56	0.80	1

*: Negative Strong Correlation and significant. **: Positive Moderate Correlation and significant.

## Data Availability

Data took from GADSM.org, Openstreetmap, Worldpop, PoI Google.

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
