# Peer review of "The Evaluation of Geographical Health Facilities Structure in Makassar City, Indonesia"

_ijerph, 2023, doi:10.3390/ijerph20065210_

Round 1
Reviewer 1 Report
This study evaluates the geographic structure of health facilities in Makassar City with an assessment of the effectiveness of the location of health facilities can be produced. It has clearly presented the location assessment of local health facilities with some techniques based on the ArcGIS platform. However, the whole manuscript is more like a local project report rather than a scientific article, which should show more contributions in theory or new methods. In order to improve, from my own aspect, I would suggest at least the following aspects.
1) For the Introduction part. It should put forward the key research question you wanted to solve specifically and what is the current research gap in the literature and practice. And how you structure your paper.
2) In the current literature review part, the introduction of the 'Healthy City' concept is more like a background of this research. You also need reviews of more studies to support your research methods and the detailed indicators used in your research process. More recent research is required.
3) The result part is too plain. Descriptions of the current situation are not necessary here. Instead, what is new from your research, and why. Discuss.
4) Conclusion is too short to conclude. What are your contributions to planning theory or new methods? How do your research results help in supporting local policy-making or project improvements? etc. What is the limitation of the research design and process?
Author Response
Dear Reviewer 1,
We have done revision based on your feedback. Within this message, We've attached the revised manuscript. These are response from your comments.
- We have revised the manuscript by adding some crucial information by putting the key research, the current research gap and practice
- We have adding more literatures by giving history of pandemic with impact the urban planning, 15 minutes city and also answering research gap why this study need to explore deeply.
- The result part already revised by giving more discussions.
- in the Conclusion, we have already expanded some information

Reviewer 2 Report
1. In the 4.4 part, you have discussed the correlation between different variables. However, it's a little hard to understand table 9 and table 11. If you can mark strongly correlated variables with * in the table, it will be better to understand. And it is confused about whether a larger number in the table indicates a stronger correlation between the two variables or a more significant linear relationship between the two variables. If you can explain further, it will make the article clearer.
2. You mention Covid-19 in the abstract, but you have a little discussion in the main body about that. If you can add the discussion about the relationship between the health facilities in Makassar city and Covid-19, the discussion of Covid-19 would be more reliable.
3. The article is a study of a specific city. If you can add some references or examples, like the evaluation of health facilities in other places, and point out what is the innovations in your study, the article will be more complete.
Author Response
Dear Reviewer 2,
Herewith message, we attached the revised manuscript. Also we respond your feedback:
- The part result and discussions have already conducted major revision by adding more analysis and discussions.
- We have added more information and correlation between the structure of Makassar city, health facilities and COVID-19
- We have added some information by comparing some areas at the literature review chapter

Round 2
Reviewer 1 Report
The revised version has improved a lot.